# Structure of Type-I *Mycobacterium tuberculosis* fatty acid synthase at 3.3 Å resolution

Nadav Elad[1], Szilvia Baron [ID] [2], Yoav Peleg[3], Shira Albeck[3], Jacob Grunwald[3], Gal Raviv[2], Zippora Shakked[2], Oren Zimhony[4] & Ron Diskin [ID] [2]

Tuberculosis (TB) is a devastating and rapidly spreading disease caused by *Mycobacterium tuberculosis (Mtb)*. Therapy requires prolonged treatment with a combination of multiple agents and interruptions in the treatment regimen result in emergence and spread of multi-drug resistant (MDR) *Mtb* strains. MDR *Mtb* poses a significant global health problem, calling for urgent development of novel drugs to combat TB. Here, we report the 3.3 Å resolution structure of the ~2 MDa type-I fatty acid synthase (FAS-I) from *Mtb*, determined by single particle cryo-EM. *Mtb* FAS-I is an essential enzymatic complex that contributes to the virulence of *Mtb*, and thus a prime target for anti-TB drugs. The structural information for *Mtb* FAS-I we have obtained enables computer-based drug discovery approaches, and the resolution achieved by cryo-EM is sufficient for elucidating inhibition mechanisms by putative small molecular weight inhibitors.

[1] Department of Chemical Research Support, Weizmann Institute of Science, Rehovot 7610001, Israel. [2] Department of Structural Biology, Weizmann Institute of Science, Rehovot 7610001, Israel. [3] Structural Proteomics Unit, Life Sciences Core Facilities, Weizmann Institute of Science, Rehovot 7610001, Israel. [4] Kaplan Medical Center, Rehovot, affiliated to the School of Medicine, Hebrew University, Jerusalem 7661041, Israel. These authors contributed equally: Nadav Elad, Szilvia Baron. Correspondence and requests for materials should be addressed to O.Z. (email: Oren.Zimhony@weizmann.ac.il) or to R.D. (email: ron.diskin@weizmann.ac.il)

Tuberculosis (TB) is becoming a major worldwide health problem in recent years. The World Health Organization estimates that in 2016 alone 10 million people contracted TB and about 1.7 million people died from the disease, mostly in low and mid-income countries[1]. TB is caused by *Mycobacterium tuberculosis (Mtb)* and for curing TB, a prolonged treatment with a combination of multiple agents is required[1]. The multiple drugs needed for a cure may be out of reach for people in parts of the world. Furthermore, the lengthy regimen leads to adherence problems and undesired interruptions that subsequently drive the emergence and spread of MDR *Mtb* strains. Novel drugs that could target MDR *Mtb* are urgently needed to combat TB.

One of the fundamental aspects of mycobacteria that confers resistance to antimicrobial agents and promotes the virulence of *Mtb* is a thick layer of mycolic acids that constitute their cell wall[2,3]. Mycolic acids are $C_{74}$–$C_{90}$ fatty acids that are synthesized by a unique combination of two canonical enzymatic systems that *Mtb* has: a fatty acid synthase-II (FAS-II) that is comprised of a group of discrete enzymes, a system that is typically found in plants and prokaryotes. The second system is the mycobacterial FAS-I, which is a multi-domain, multi-functional enzyme complex that is typically found in fungi and higher eukaryotes[4,5] but not in prokaryotes. The FAS-I complex of mycobacteria is an essential enzymatic complex[6], making it an attractive drug target. The mycobacterial FAS-I has a distinct capacity to elongate fatty acids beyond $C_{16}$, to $C_{24/26}$[7,8], which are further elongated to meromycolate ($C_{56}$) by the FAS-II system and then condensed with $C_{26}$ to form mycolic acids[2,5,9]. A group of compounds that interfere with fatty acid synthesis by *Mtb* have become invaluable drugs for fighting TB. Specifically, isoniazid and ethionamide, first and second line drugs against TB, inhibit mycolic acid synthesis through inhibition of an enoyl reductase (ER) of the FAS-II system[10]. Pyrazinamide (PZA) is a first line and an indispensable drug for shortening the regimen required for curing TB[11]. Analogs of PZA were shown to bind and inhibit mycobacterial FAS-I[12,13] by a mechanism that is still unknown.

Here we report the 3.3 Å resolution structure of the ~2 MDa FAS-I from *Mtb*, determined by single particle cryo-EM. The *Mtb* FAS-I structure reveals unique adaptations at the catalytic modules that allow production of long fatty acids. The structural information for *Mtb* FAS-I we have obtained makes an important milestone in the quest for novel drugs as it now enables computer-based drug discovery approaches. Importantly, the resolution achieved by cryo-EM is sufficient for elucidating inhibition mechanisms by putative small molecular weight inhibitors. Differences in the catalytic domains of *Mtb* FAS-I compared to a structurally similar fungal FAS-I homolog suggest a possibility to target *Mtb* FAS-I with high specificity that would be needed to minimize undesired toxicity toward other systems like the human FAS-I.

## Results

**Determining the *Mtb* FAS-I structure to near atomic resolution.** *Mtb* FAS-I is an α6 subtype complex, composed of six long polypeptide chains with an overall molecular weight of ~2 MDa. Each of the α chains has seven catalytic domains and is 3069 amino acids long. Structural information at near atomic resolution is available for FAS-I from fungi[14–16]. The FAS-I system from the non-pathogenic mycobacterium *M. smegmatis* was investigated using cryo-electron microscopy (cryo-EM), yielding a 7.5 Å resolution structure[17]. The FAS-I system from *Mtb* itself was also studied using cryo-EM, but a structure was obtained to only 20 Å resolution due to conformational flexibility of the purified FAS-I complex[18]. The lack of structural information at near atomic resolution for the *Mtb* FAS-I system hinders the

ability to investigate the mechanism by which PZA or its analogs inhibit FAS-I, and precludes rational design approaches for developing novel inhibitors as potential TB therapeutics. To enable structural studies of *Mtb* FAS-I we developed a platform for producing the activated form of the protein complex[19]. This platform relies on co-expressing *Mtb* FAS-I in *E. coli* together with 4′-phosphopantetheinyl transferase (AcpS) protein, which activates the acyl carrier protein (ACP) module of FAS-I. Following gentle purification, homogeneous FAS-I complex can be clearly seen using negative-stain EM (Supplementary Fig. 1), as well as using cryo-EM, which yields well-defined 2D classes (Fig. 1a). A 3D map with D3 symmetry imposed was reconstructed from a final data set of 40,160 particles at 3.3 Å overall resolution with good angular coverage (Supplementary Fig. 2a, b). The map reveals a barrel shaped complex (Fig. 1b) as previously observed for fungal and mycobacterial FAS-I[14,15,17,18,20,21]. 3D classification indicated that the core barrel domains had a uniform conformation, and conformational heterogeneity was observed only for the ACP modules (see below). The local resolution of the map extends beyond 3.0 Å at core regions of the catalytic subunits and decreases to ~6 Å at the more peripheral and flexible regions (Fig. 1b, Supplementary Fig. 3). This map allowed us to model the structure of FAS-I including unambiguous assignment of side chain rotamers for most amino acids at the better-resolved regions (Fig. 1c, Supplementary Fig. 4).

**ACP adopts variable orientations at the vicinity of KS.** Out of seven functional domains of *Mtb* FAS-I, we modeled the six domains that make the barrel-like cage (Fig. 2a, Supplementary Table 1). The density for the ACP was weak and thus we did not model it (see Methods section). The function of ACP is to carry the growing fatty acid at the inner chamber between the catalytic domains[21,22] (Supplementary Fig. 5). The weak density for the ACP implied that FAS-I particles are heterogeneous with respect to the locations of the six ACP domains at the inner chamber. To gain insights for the putative location of ACP we classified the data set into five distinct 3D classes and refined the maps separately with imposed D3 symmetry. Clear densities for the ACP domains are visible in all the five different classes at resolutions of about 15 Å (Fig. 2b, Supplementary Fig. 6). In all classes, the ACP domains are near the ketoacyl synthase (KS) modules in the central wheel as was previously observed for the yeast FAS-I[15,21]. However, the exact orientations (and likely conformations) of the ACP domains vary between the different classes (Supplementary Fig. 6), explaining the weak density in the original reconstruction that included all particles. Imposing D3 symmetry for this analysis may obscured low populated states in which some of the ACP domains are located at different positions in the catalytic chamber, as was previously observed in a fungal FAS-I[22]. We therefore used the localized reconstruction method[23] to extract individual asymmetric units (referred to as sub-particles) from the 2D particle images, each focusing on a single ACP domain. After excluding sub-particles that overlap in the 2D images, we obtained a data set of 166,918 sub-particles. We then classified the sub-particle data set in 3D into 3, 5, or 10 classes (Supplementary Fig. 7). The localized reconstruction revealed orientations of the ACP which were not observed in the context of the whole complex, however the resolution of the ACP within the sub-particle classes was similar to that obtained using 3D classification of the whole complex. This indicates that the ACP domains are not rigidly held in a single position within the chamber but rather occupy a continuum of closely related orientations and possibly conformations. Interestingly, in all classes the ACP domain remains localized at the vicinity of the KS module.

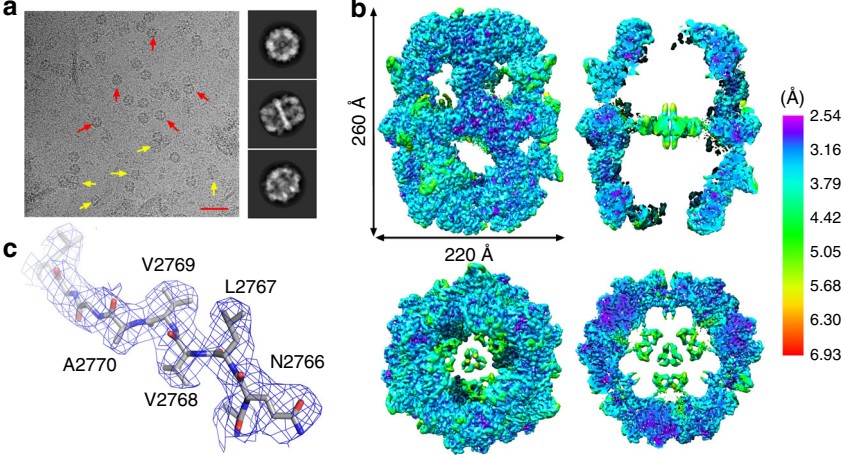

**Fig. 1** Determining a 3.3 Å resolution map using single particle cryo-EM. **a** Cryo-EM micrograph of *Mtb* FAS-I particles. FAS-I particles are clearly visible in the micrographs. Scale bar represents 50 nm. Red and yellow arrows highlight some particles with 'top' or 'side' views, respectively. Representative 2D-class averages of 'top', 'side', and 'tilted' views are shown on the right side of the micrograph from top to bottom, respectively. **b** Electron density map of FAS-I colored by the local resolution estimate. Resolution lower than 3.0 Å is colored purple/blue, between 3.0 and 5.0 Å is colored cyan/green and higher than 5.0 Å is colored in yellow/red. 'Side' views (upper images) and 'top' views (lower images) of the density map are shown. The panels on the right are cross-sections of the map, illustrating the quality of density in the central regions of the map. **c** A segment spanning residues 2765–2773 of FAS-I illustrating the map quality at inner regions at better than 3.0 Å resolution

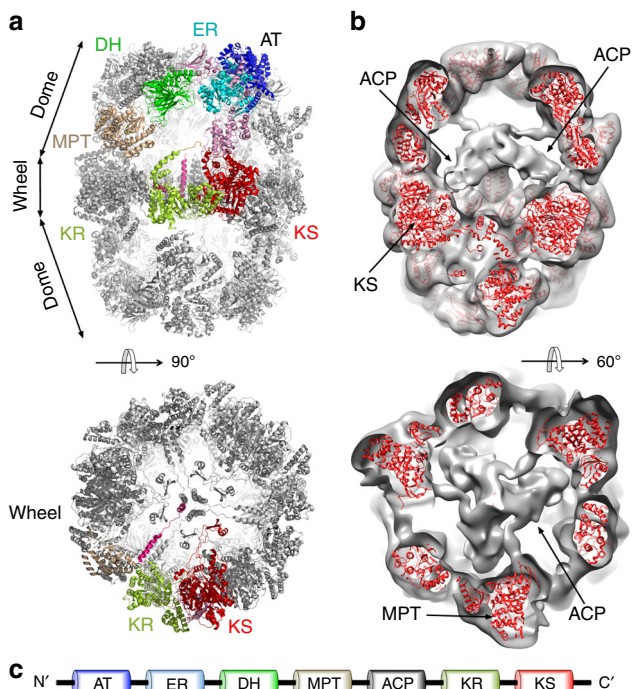

**Fig. 2** Structure of the *Mtb* FAS-I. **a** The final model of Mtb FAS-I is shown using ribbon representation. A 'side' view (upper image) reveals the barrel-like shape of FAS-I. One of the six chains that compose the D3-symmetric barrel is colored. Each of the six different catalytic domains that make the barrel is shown using a distinct color: acetyltransferase (AT) in blue, enoyl reductase (ER) in cyan, dehydratase (DH) in green, malonyl transacylase (MPT) in brown, ketoacyl reductase (KR) in light green, and ketoacyl synthase (KS) in red. A 'top' view (lower image) shows the wheel-like organization of the KR and KS domains. **b** The *Mtb* FAS-I model as red ribbon is shown inside a low-pass filtered map from one specific 3D class as 'tilted' and 'top' views (upper and lower images, respectively). A density for the acyl carrier protein (ACP) is visible. **c** A schematic diagram showing the domain organization of *Mtb* FAS-I. Color coding and abbreviations are the same as in **a**

**Inverted surface electrostatic potentials of the *Mtb* FAS-I**. Putative docking of ACP domains to the KS modules is partially mediated by a long loop (residues 2210–2226) that projects from the neighboring ketoacyl reductase (KR) module (KR-loop) (Supplementary Fig. 8). An equivalent such loop is not seen in the case of FAS-I from fungi. It was previously noted that electrostatic interactions govern the interaction of the negatively charged ACP with the positively charged catalytic modules in *S. cerevisiae* FAS-I[15]. Examining the surface electrostatic potential near the active site of the KS module from *Mtb* FAS-I reveals mixed charged residues (Fig. 3a) in contrast to the positively charged surface seen in FAS-I from the thermophilic *T. lanuginosus* fungus (PDB: 4v59)[14] (Fig. 3b). However, the KR-loop that mediates the interaction of ACP with KS (Supplementary Fig. 8) donates negatively charged residues (Fig. 3c) and ultimately makes a negatively charged surface near the active site of KS (Fig. 3d). Negatively charged surfaces are also found near the catalytic sites of the acetyltransferase (AT), ER, dehydratase (DH) (Supplementary Fig. 9), and KR modules (Supplementary Fig. 10), suggesting that in the case of *Mtb* FAS-I electrostatic interactions may also mediate attachment of ACP to the catalytic modules but with an opposite charge as in the fungal FAS-I system. Compared to the other catalytic modules, MPT has a more positively charged surface near its catalytic site, which might lower the probability of ACP to interact with it during the transport of the growing fatty acid between the catalytic modules. Since MPT has a special role in terminating the synthesis cycle (Supplementary Fig. 5), its positively charged surface may function as a safety mechanism that prevents premature termination of synthesis. Since ACP needs to shuttle malonyl from MPT to KS during synthesis, it could be that the electrostatic potential of ACP changes when it binds the growing aliphatic chain. Such a change might be induced by the burial of the aliphatic chain of the fatty acid at the core of the ACP as was previously suggested to happen during substrate shuttling[21]. Such electrostatic modulations of the ACP are likely to be transient as the aliphatic chain is only transiently concealed by the ACP[24].

**Large inner cavities at the catalytic modules of the *Mtb* FAS-I**. In addition to differences in electrostatic potentials, comparing

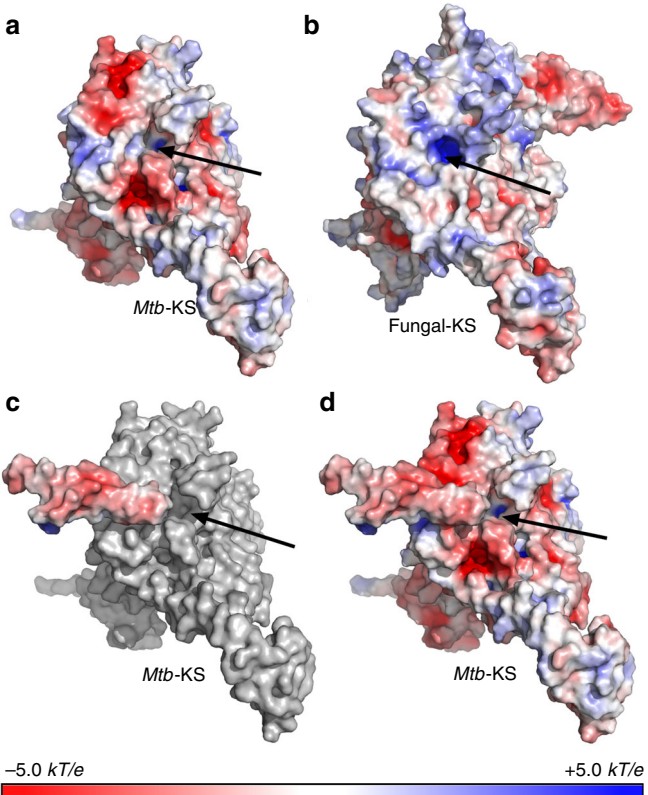

**Fig. 3** A KR-derived loop provides negatively charged surface to KS. **a** Electrostatic potential at the solvent accessible surface mapped on the molecular surface of the KS module from *Mtb* FAS-I. Negatively charged areas are in red ($-5.0$ $kT/e$) and positively charged areas are in blue ($+5.0$ $kT/e$). Electrostatic potentials were calculated using APBS tools[36]. The catalytic site is indicated with a black arrow. **b** Electrostatic potential of the KS module from fungal FAS-I (PDB: 4v59). **c** Electrostatic potential of the loop donated from the *Mtb*-KR module on top of a surface representation of the KS module in gray. **d** Combined electrostatic potentials of the KS module with the KR-originating loop. The surface near the active site of KS is dominated by negative charge in the case of *Mtb* and by positive charge in the case of the fungal FAS-I

*Mtb* FAS-I to the fungal FAS-I (PDB: 4v59)[14], reveals a marked structural difference in the catalytic clefts of some of the active modules (Fig. 4). The fungal FAS-I is an homolog that retains a barrel shape complex and that structural information at near-atomic resolution is available for. The catalytic cleft of the ER from *Mtb* FAS-I is wider compared to the fungal-ER, and the flavin mononucleotide (FMN) co-factor in *Mtb* FAS-I is more exposed compared to the FMN in the fungal FAS-I. This difference is mostly due to the local amino acid composition rather than changes in backbone conformation (Fig. 4, Supplementary Movies 1 and 2). In this regard, the quality of density in the inner region of the protein (Fig. 1b, c) is sufficient to accurately model side-chain rotamers (Supplementary Fig. 11) that ultimately determine the shape of the cleft. The catalytic cleft of the KR module is also much wider in *Mtb* FAS-I compared to fungal FAS-I (Fig. 4, Supplementary Movies 3 and 4). A notable difference that contributes to this change is a distinct backbone conformation of a loop that partially conceals the active site (Fig. 4). A similar difference in width is also observed between the DH modules of *Mtb* and the fungal FAS-I (Fig. 4, Supplementary Movies 5 and 6). More striking is the larger inner volume of the

*Mtb* DH catalytic chamber compared to the fungal protein (Supplementary Fig. 12). In contrast to these observations, the catalytic pockets of the KS, MPT, and AT modules (Fig. 4, Supplementary Movies 7 and 8) have similar sizes in *Mtb* compared to the fungal FAS-I (Supplementary Fig. 13). The extended catalytic clefts of the KR, DH, and ER modules may reflect evolutionary adaptation of *Mtb* FAS-I for producing the unusually long $C_{26}$ fatty acids; larger clefts are probably required for accommodating the longer aliphatic chains, while the ACP-attached pantothenic acid reaches into the active sites. As for the other catalytic modules, the KS and MPT from both species seem to have long channels or clefts that effectively do not restrict the length of the growing aliphatic chain (Supplementary Fig. 14), and the AT module transfers similarly sized acetyl groups (Supplementary Fig. 5) in both species.

## Discussion

Despite overall similar architecture, *Mtb* FAS-I has distinct features that differentiate it from previously determined fungal FAS-I systems. The transition from positively charged surfaces of the fungal FAS-I catalytic modules to negatively charged surfaces in *Mtb* FAS-I and the likely complementary change in the charge of the ACP reflect the large evolutionary distance between mycobacteria and fungi. The unique ability of *Mtb* FAS-I to produce $C_{26}$ fatty acids can explain its larger catalytic clefts in the KR, DH, and ER modifying modules as compared with the fungal FAS-I. In spite of the evolutionarily conserved biochemical process of fatty acid synthesis, the combination of large catalytic clefts together with the altered electrostatic potentials near the active sites, makes the catalytic modules of *Mtb* FAS-I unique. Considering that targeting mycolate synthesis is a proven approach for fighting TB[9], inhibiting *Mtb* FAS-I is a promising strategy for fighting MDR-*Mtb*. The structural data that we provide here can now be used for virtual screening approaches[25] for identifying novel drug candidates. Moreover, the near atomic resolution that we achieved in this study using only a modest number of particles suggests that it would be possible to further use structure-guided approaches to improve putative hits from such screening efforts. Along these lines, elucidating the inhibition mechanism of PZA analogs and rationally improving them using structural data should also be possible on the base of single particle cryo-EM studies.

## Methods

**Protein expression and purification.** To produce FAS-I for cryo-EM analysis, we followed a protocol that we previously developed[19]. Briefly, FAS-I expression plasmid (pMT100 StrepFlag-FAS-I) was transformed into *E. coli* BL21 (DE3) (Novagen) cells together with AcpS expression plasmid (pACYCDuetAra-Acps). Expression of AcpS was initiated by adding 0.2% Arabinose followed by the induction of FAS-I expression by adding 200 μM Isopropyl β-D-1-thiogalactopyranoside at 15 °C for 16–18 h. Cells were lysed using French press in buffer 'A' (100 mM potassium phosphate buffer (KPB) pH 7.2, 150 mM potassium chloride, 1 mM tris(2-carboxyethyl)phosphine, 1 mM ethylenediaminetetraacetic acid) supplemented by protease inhibitor cocktail (cOmplete, Roche) and 20% sucrose. After clarifying the supernatant by 1 h centrifugation at 150,000×*g*, protein debris were pelleted by precipitation at 20% (w/v) ammonium sulfate. Next, FAS-I was precipitated by adjusting the ammonium sulfate concentration to 50% (w/v) and applying 150,000×*g* for 1 h. Precipitate was dissolved in buffer 'A' supplemented by protease inhibitor cocktail (cOmplete, Roche) without sucrose, and FAS-I was captured using Strep-Tactin Sepharose column (GE Healthcare) and separated by FPLC system. Purified FAS-I was eluted using buffer 'A' supplemented with 2.5 mM d-desthiobiotin.

**Negative staining.** 3.5 μl of the purified FAS1 sample at 0.4 mg/ml was applied to glow-discharged, homemade 300 mesh carbon-coated copper TEM grids for 30 s. Excess liquid was blotted and, after a wash with distilled water, the grids were stained with 2% uranyl acetate. Samples were visualized in an FEI Tecnai T12 TEM operated at 120 kV, equipped with a Gatan OneView camera.

**Cryo-EM image acquisition.** 3.5 μl of FAS-I solution at 10 mg/ml was applied to glow-discharged C-flat 2/2, 200 mesh holey carbon grids and plunge frozen in

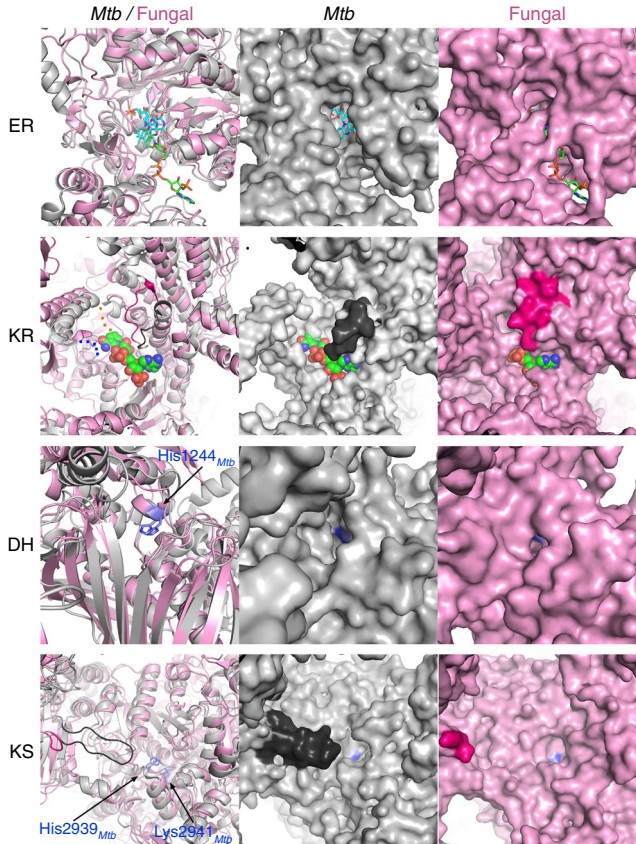

**Fig. 4** Catalytic clefts of *Mtb* FAS-I structurally differ from fungal FAS-I. The four catalytic domains that take part in extension and modification of the growing aliphatic chain during fatty acid synthesis are shown using ribbon representation (left column). The polypeptide chains of *Mtb* FAS-I (gray) and of the fungal FAS-I (pink, PDB code: 4V59) are superimposed. Surface representations of *Mtb* FAS-I (middle column) and of fungal FAS-I (right column) illustrating the differences in shapes of the catalytic clefts between *Mtb* and fungal FAS-I. The enoyl reductase domains (ER, top row) are shown with the bound FMN co-factors in cyan. The fungal FAS-I structure was determined with a bound NADPH that is colored green. In the ketoacyl reductase domain (KR, second row from top) of *Mtb* FAS-I, two short loops were not visible in the density maps (blue and orange dots). NADPH from the structure of the fungal FAS-I is shown in green space-filling spheres. A loop that partially conceals the catalytic cleft in the fungal FAS-I (hot pink, residues 750–761) adopts a different structure in *Mtb* FAS-I (black, residues 2237–2248). The NADPH from the fungal FAS-I is shown in the surface representation of *Mtb*-KR to help visualize the catalytic cleft. The catalytic histidine residues of the dehydratase domains (DH, third row from top) are colored blue. The catalytic lysine and histidine residues of the ketoacyl synthase domains (KS, lower row) are colored blue. A long loop that projects from *Mtb*-KR toward the active site of *Mtb*-KS is shown in black. The corresponding loop in the fungal FAS-I is colored in hot pink. For better 3D perception, movies of the various catalytic modules are included (Supplementary Movies 1, 2, 3, 4, 5, 6, 7 and 8)

liquid ethane cooled by liquid nitrogen, using a Leica EM-GP plunger (3 s blotting time, 80% humidity). Grids were screened on an FEI F20 microscope equipped with a K2 Summit direct detector (Gatan). Cryo-EM data were collected on a Titan Krios electron microscope (FEI) operated at 300 kV. Coma-free alignment was performed with AutoCTF (FEI) and beam size was 645 nm. Movies were recorded on a K2 Summit direct detector mounted at the end of a GIF Quantum energy filter (Gatan) using a slit of 20 eV. 5248 Movies were collected in super-resolution counting mode at a nominal magnification of 130,000× corresponding to a physical pixel size of 1.054 Å. The dose rate was set to 4.49 electrons per physical pixel per second and the total exposure time was 20 s, resulting in an accumulated dose

of 80.1 electrons per Å². Each Movie was fractionated into 40 frames of 0.5 s. Nominal defocus range was −0.5 to −2.5 μm. All dose-fractionated images were recorded using an automated low dose procedure implemented in SerialEM[26]. Stage navigation was used to navigate to hole centers and image shift used to target 5 distinct imaging locations within each hole.

**Single particle cryo-EM image processing.** Image frames were Fourier-binned 2 × 2, aligned (5 × 5 tiles, B-factor 100) and dose-weighted using MotionCor2[27]. Contrast transfer function (CTF) parameters were estimated using Gctf[28] on the non-dose-weighted image sums. 1912 images showing resolution better than 6.0 Å were selected for further processing. All subsequent image processing was done using RELION 2.0[29]. About 500 particles were manually picked and subjected to reference-free 2D classification, followed by automated picking using the newly generated 2D class averages. 91,589 automatically picked particles, extracted into 400² pixel boxes and subjected to multiple rounds of 2D classification in order to clean the data set. 2D classes showing secondary structure features were retained at each round, resulting in a total of 40,160 particles. 3D classification with D3 symmetry imposed did not result in maps with significant difference at the core elements. Additionally, separating the data set into different numbers of 3D classes resulted in lower resolution in the classes. Therefore all 40,160 were subjected to 3D refinement with a sphere mask of 290 Å and D3 symmetry imposed, using the published cryo-EM structure of FAS-I from *M. smegmatis*[17] (PDB: 4V8L) filtered to 60 Å as an initial model. Only movie frames 2–15 were used at this stage. The global resolution in the refined structure, estimated by the gold-standard FSC = 0.143 criteria, was 3.3 Å (Supplementary Fig. 2a). Local filtering for subsequent model building was done using RELION, and local resolution estimate was calculated using the program blocres in BSOFT[30].

Localized reconstruction was performed as described in Ilca S.L. et al.[23] using scripts (version 1.2) embedded in Scipion[31] and using RELION 2.1 for 3D classification and reconstruction. The position of a single ACP domain and the center of the complex were marked in the refined, D3 symmetrized map using UCSF Chimera[32]. These coordinates were then used to locate all symmetry-related positions of the ACP domains in the 2D particle images. A minimum distance of 25 pixels was set between extracted ACP coordinates to avoid overlaps. A total of 166,918 ACP domains ('sub-particles') were extracted (160² pixel boxes). The sub-particle data set was 3D classified into 3, 5 and 10 classes using the alignment and orientation parameters obtained for the whole complex and a spherical mask of 120 Å diameter. Local resolution estimate of some sub-particle 3D maps indicated regions where the ACP was resolved better than 10 Å, however other ACP regions remained noisy at that resolution. Therefore, to avoid overfitting we low-pass filtered all classes to 15 Å.

**Model building and refinement.** The 4V8V PDB entry was used as an initial model. First, it was docked into the electron density map using UCSF Chimera[32], and then it was real-space refined using Phenix[33]. The model was then manually adjusted to fit the electron density map while converting the amino acids to correspond to the *Mtb* FAS-I sequence using Coot[34]. Subsequently, we used iterative real-space refinements and manual model improvements using Phenix and Coot, respectively. The final model spans residues 32–3061. We did not observe interpretable density for the first 31 residues, the last 34 residues, or for residues 2289–2296 and 2350–2354 and hence did not model them. Also, density for the ACP domain (residues 1742–1976) was weak, and hence it was not modeled.

**Structural analysis and visualization.** To visualize structures, electron density maps and to perform structural analysis we mainly used PyMol[35] and UCSF Chimera[32]. Electrostatic potentials were calculated using APBS tools[36]. To detect and visualize internal cavities we used the cavity detection module of PyMol with cavity detection cutoff of 2 solvent radii and cavity detection radius of 10 solvent radii.

## Data availability

Atomic model for the *Mtb* FAS-I as well as EM density maps are available at the protein data bank under accession code 6GJC and at the electron microscopy data bank under accession code EMD-0011. Other data are available from the corresponding authors upon reasonable request.

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

## Acknowledgements

We thank Wim Hagen from EMBL Heidelberg for his invaluable assistance in collecting cryo-EM data. We thank Prof. Deborah Fass for critical comments and suggestions. This project was continuously supported by the Kimmelman Center for Biomolecular Structure and Assembly. O.Z. was supported by the Legacy Heritage Clinical Research Initiative of the Israel Science Foundation (Grant no. 1629/10) and by the Israeli Ministry of Health Chief Scientist award (Grant no. 6223). S.B. was supported by the Israeli Minister of Absorption for Post Doctorate Fellowship no. 17733. Electron microscopy studies were supported in part by the Irving and Cherna Moskowitz Center for Nano and Bio-Nano Imaging at the Weizmann Institute of Science. High-resolution cryo-EM data were acquired at the cryo-EM facility of the European Molecular Biology Laboratory (EMBL) in Heidelberg supported by iNEXT (Project no. 1975) and funded by the Horizon 2020 program of the European Union. Research in the Diskin lab is supported by a research grant from the Enoch Foundation, a research grant from the Abramson Family Center for Young Scientists, a research grant from Ms. Rudolfine Steindling, by the Minerva Foundation with funding from the Federal German Ministry for Education and Research, and by a grant from the Israel Science Foundation (Grant no. 682/16). R. D. is an incumbent of the Tauro career development chair in biomedical research.

## Author contributions

O.Z., Z.S. and R.D. designed the project. O.Z. oversaw the project. Y.P. developed the expression vector. S.A., S.B. and J.G. developed protein production, purification, and validation methods. S.B. and G.R. produced and purified protein. N.E. and S.B. performed initial EM characterization of the protein. N.E., S.B. and R.D. collected cryo-EM data. N.E. and R.D. processed cryo-EM data, calculated density maps, and modeled the structure. R.D. performed structural analysis and wrote the manuscript with the help of all other authors.

## Additional information

**Competing interests:** The authors declare no competing interests.

