## [Peer Review File · Nature Communications]

Reviewers' comments:

Reviewer #1 (Remarks to the Author):

In their manuscript, Elad et al. report a structure of mycobacterial fatty acid synthase (FAS-I) determined by single-particle cryo electron microscopy at 3.3Å resolution. Mycobacterial FAS-I forms a large, D3 symmetry assembly, which is generally conserved between bacterial and fungal Type 1 FAS. Several related FAS assemblies have been visualized at similar resolution, but for mycobacterial FAS-I only substantially lower resolution data, which don't allow atomistic modelling, have been reported previously. The provided structure is principally of high quality, but the manuscript is of a particularly descriptive nature. It focusses on unique aspects of mycobacterial FAS-I and highlights details of active site surroundings and tunnels possibly relevant for the particular product range of mycobacterial FAS-I. The analysis of ACP localization and interpretation of ACP linkage suffers from methods limitations. Hypotheses on the relevance of particular features are not validated by other methods. Data directly supporting or validating use of this system in drug discovery are not reported. To increase the relevance of this manuscript, the authors should attempt resolving a bound pharmacological inhibitor of mycobacterial FAS-I, or to provide conclusive data on ACP localization or substrate interactions at a resolution that at least resolves individual secondary structure elements in ACP. For detailed comments, see below.

Detailed comments:

Intro

Page 2, end of first paragraph: Are difference to fungal FAS-I the most relevant feature for minimizing undesired toxicity ?

Page 2, last line: elongate

Results

Page 4; ACP-connectivity: The authors have decided, likely due to the low overall number of particles, to enforce D3 symmetry also for the analysis of ACP positions. The fact that ACPs are resolved only at very low resolution, indicates that they occupy multiple positions, have enhanced flexibility at one position, or do simply not obey D3 symmetry. The very low resolution visualization does not provide any recognizable features, e.g. secondary structure elements. The authors mention that ACP is close to KS active sites, but their position is not qualified by directly being bound to KS with measurable KS-Cys to ACP-Ser-PPant distances. The reported averaged density might well be affected by artefacts from averaging across multiple non-identical states and should not be used to discuss connectivity or interactions of ACP domains. To obtain trustworthy interpretations, the authors would need to release symmetry constraints, possibly requiring a larger data set, and still be able to resolve at least secondary structure elements of ACP.

Page 5; MPT and electrostatic potential in catalysis: In related fungal systems, substrate concentrations have a profound effect on product spectra. To support their hypothesis on the relevance of specific features of mycobacterial FAS-I did the authors considers to do product assays ?

Page 5; ACP interactions and aliphatic chain burial: Consider adding a reference to DOI: 10.1021/bi5014563

Page 6, 7: The authors suggest that using data collected in 2.5 days and processed in a few days/2 weeks, their preparations of mycobacterial FAS-I with added commercial inhibitor should be suited to resolve structural details of inhibitor interactions. This indeed appears to be a feasible and relevant experiment, and could be included in this manuscript to increase the overall

relevance of the work.

Figures:

Discussion of ACP linkage and orientation would require at least one figure with fitted ACP that allows assessment of relative orientation of ACP and proximal enzymatic domains, e.g. KS.

Using a consistent colour scheme for domains across figures would be helpful.

Ext. Data Fig. 2a: Please provide the full set of FSC curves produced by Relion.

Ext. Data Fig. 4: Interpretation of central averaged density as individual ACP orientations would require improved data. It is difficult to exclude that the current maps still provide an ensemble average of multiple ACP positions.

Ext. Data Fig. 6: Indicate KS active site and a measurement of length of the path to the most plausible/possible PPant attachment point on ACP.

Ext. Data Fig. 8/9: Provide full methods for electrostatic potential calculation and a colour scale together with the figures.

Ext. Data Fig. 11,13: Provide methods for tunnel calculation

Reviewer #2 (Remarks to the Author):

The manuscript by Elad et al. describes the structure of the 2 MDa multienzyme complex fatty acid synthase (FAS) from *Mycobacterium tuberculosis* determined by cryo-electron microscopy. FAS is responsible for making the long fatty acids that constitute the mycobacterial cell wall, an important factor in virulence of *M. tuberculosis*. The complex was expressed in *E. coli* together with the phosphopantetheinyl transferase that is necessary to activate the acyl carrier protein (ACP) domain that carries the growing fatty acid chain between the different catalytic sites. The resulting preparation yielded a cryo-EM map with a resolution of 3.3 Å, showing a rigid structure with heterogeneity only for the ACP inside the barrel. 3D classification revealed the ACP mostly docked at the ketoacyl synthase domain, as is also the case for eukaryotic FAS structures. Compared to the fungal FAS systems, the authors find larger catalytic clefts for some enzymes, compatible with the longer fatty acids produced by mycobacterial FAS. In addition, the authors observe a reversal of the electrostatic potentials of the catalytic domains.

The paper shows the first atomic resolution structure of mycobacterial FAS, an enzyme important for *M. tuberculosis* virulence and a potential drug target. For this reason, it will be of high interest in the field. The EM work is of high technical quality and the paper is overall well written; however, the presentation of the data needs some attention. The quality of the cryo-EM map is not shown clearly in any of the figures, so that the resolution claims can not be judged. In addition, details of the structure are difficult to interpret in still images and it would be useful to add some movies.

Specific points:

1. The last paragraph of the introduction describes the features and importance of the mycobacterial FAS system. It seems more logical to put this material before the section that describes the details of the structure.

2. p. 4 first paragraph. The authors claim a resolution of "almost 2.5 Å" for parts of the structure. Such a claim should be supported by figures showing the relevant map regions with fitted model.

At this resolution, most rotamers are unambiguous and water molecules will be visible. However, figure 1b does not really indicate much density beyond 3 Å. This means that the "unambiguous assignment of side chain rotamers for the vast majority of the amino acids" is also probably an overstatement. A supplementary figure with details for more regions than the very short helix in figure 1c should be included. (Resolution claims below 3 Å would be supported by side chains of isoleucines, and an obvious difference between phenylalanines and tyrosines.) A panel for the active site of each catalytic domain would be preferable. Also figure 1c should be replaced with a better example. The helix shown contains several methionines and their densities can't be judged from the angle that is shown.

3. p. 4, ACP classification: seeing multiple orientations of the ACP docked on the KS domain is interesting, but it should not be overinterpreted. The classification was done with D3 symmetry applied, but there is no reason to assume that the six ACP domains in a complex, or even the three in one dome, are communicating. This means that the observed structures are probably averages of different conformations, as also suggested by the low resolution of ~15 Å. However, the authors even claim to see the linker between the ACP and the MPT domain in one of the classes. Even if it was rigid, such a feature would not be visible at 15 Å resolution. The interpretation of this linker, including figure 2b and extended data figure 5, should be removed.

4. p. 4, last paragraph: it is not clear if the KR loop is unique for mycobacteria or is also present in fungal FAS. This is relevant for the discussion of electrostatic surfaces.

5. p. 4/5 and Ext. data figure 7: the header of the chapter is "inverted potentials". But the figure (b vs. d) suggests local changes from positive to neutral and from neutral to negative, which means overall more negative charges, but locally not inverted. In both species, the area above the active site (arrow in b and d) is positive and below it negative.

6. p. 5 bottom: "a marked diversity in the catalytic clefts of some modules". As only two species are considered, the authors probably mean "a marked difference". A similar issue, top of p. 6: "variations in amino acid composition".

7. Figure 3: it is very hard to understand this figure, especially the surface representations. Movies or stereo figures would be better to give a 3D impression.

8. Extended data figure 5: this figure is impossible to interpret and would be better shown as a movie or at least a stereo figure. But as stated in point 3 above, the conclusion about the connectivity of ACP and KS is not supported by data.

9. Extended data figure 6: Also this figure is hard to interpret, in particular it is not clear how the KR loop model relates to the density map, and how it inserts between ACP and KS.

10. Table 1: Model to map correlation 0.8282: it is not clear what this one figure means. It would be better to add a map-to-model FSC curve to figure 2a. The map-to-model FSC 0.5 resolution should be similar to the half-map resolution at FSC 0.143.

11. Table 1: please add MolProbity score and clashscore.

Point-by-point response to reviewers

Reviewers' comments:

Reviewer #1 (Remarks to the Author):

In their manuscript, Elad et al. report a structure of mycobacterial fatty acid synthase (FAS-I) determined by single-particle cryo electron microscopy at 3.3Å resolution. Mycobacterial FAS-I forms a large, D3 symmetry assembly, which is generally conserved between bacterial and fungal Type 1 FAS. Several related FAS assemblies have been visualized at similar resolution, but for mycobacterial FAS-I only substantially lower resolution data, which don't allow atomistic modelling, have been reported previously. The provided structure is principally of high quality, but the manuscript is of a particularly descriptive nature. It focusses on unique aspects of mycobacterial FAS-I and highlights details of active site surroundings and tunnels possibly relevant for the particular product range of mycobacterial FAS-I. The analysis of ACP localization and interpretation of ACP linkage suffers from methods limitations. Hypotheses on the relevance of particular features are not validated by other methods. Data directly supporting or validating use of this system in drug discovery are not reported. To increase the relevance of this manuscript, the authors should attempt resolving a bound pharmacological inhibitor of mycobacterial FAS-I, or to provide conclusive data on ACP localization or substrate interactions at a resolution that at least resolves individual secondary structure elements in ACP. For detailed comments, see below.

Detailed comments:

Intro

Page 2, end of first paragraph: Are difference to fungal FAS-I the most relevant feature for minimizing undesired toxicity ?

From all the structurally characterized FAS-I systems, the fungal systems are the most similar to that of Mtb since they form a D3-symmetric barrel shape complexes. Hence, differences between the fungal and the Mtb systems imply that high specificity could be achieved. Obviously, the concern is to nonspecifically inhibit the mammalian systems. To better clarify this point we now rephrased the text to say: "Differences in the catalytic domains of Mtb FAS-I compared to a structurally similar fungal FAS-I homolog suggest a possibility to target Mtb FAS-I with high specificity that is needed to minimize undesired toxicity toward other systems like the human FAS-I."

Page 2, last line: elongate

We thank Reviewer #1 for spotting this typo

Results

Page 4; ACP-connectivity: The authors have decided, likely due to the low overall number of particles, to enforce D3 symmetry also for the analysis of ACP positions. The fact that ACPs are resolved only at very low resolution, indicates that they occupy multiple positions, have enhanced flexibility at one position, or do simply not obey D3 symmetry. The very low

resolution visualization does not provide any recognizable features, e.g. secondary structure elements. The authors mention that ACP is close to KS active sites, but their position is not qualified by directly being bound to KS with measurable KS-Cys to ACP-Ser-PPant distances. The reported averaged density might well be affected by artefacts from averaging across multiple non-identical states and should not be used to discuss connectivity or interactions of ACP domains. To obtain trustworthy interpretations, the authors would need to release symmetry constraints, possibly requiring a larger data set, and still be able to resolve at least secondary structure elements of ACP.

As Reviewer #1 correctly assessed, we had to imposed D3 symmetry for analyzing the ACP position. Indeed, from this analysis some uncertainty remains regarding the position of the ACP because the ACP domains do not necessarily obey D3 symmetry. Imposing of symmetry may conceal other low-populated poses of ACP, like the ones that Gipson P. et al. previously observed in the fungal system (Gipson P. et al. PNAS 2010). For this reason and based on the criticism of reviewer #1 we now added to the revised manuscript an analysis to address this point. We have used the localized reconstruction protocol that was developed by Prof. Juha Huiskonen (Ilca S. et al. Nature Communications 2015). The protocol was used to extract asymmetric portions from the FAS-I particles, resulting in ~167,000 unique sub-particles, each containing a single ACP domain. Classifying this extended set of sub-particles in 3D to different number of classes indicates that indeed the ACP domains are always localized to the vicinity of the KS module, but occupy a large number of different orientations and perhaps different conformations. Please note that we do not claim that the ACP domains are necessarily docked to the KS in a way that they can deliver substrates, although in some classes it certainly looks like the ACP are closer to the KS than in other classes (see Extended Data Figure 7). Overall, the localized reconstruction revealed many more orientations of the ACP domains, which is reminiscent to the results reported by Ilca S. et al. who applied this algorithm to other cases with flexible domains. Some of the classes provide maps for the ACP domains that have more details at slightly better resolutions. However, we could not get reconstructions at high-enough resolution that would reveal secondary structures of the ACP.

Page 5; MPT and electrostatic potential in catalysis: In related fungal systems, substrate concentrations have a profound effect on product spectra. To support their hypothesis on the relevance of specific features of mycobacterial FAS-I did the authors considers to do product assays ?

We thank Reviewer #1 for raising this interesting point. We do feel however that conducting biochemical experiments to profile the product spectra of the Mtb FAS-I system as a function of substrate concentration is beyond the scope of our structural study. Furthermore, such experiments may require expertise that we don't necessarily have. We believe that this manuscript will inspire experts in the fatty acid synthesis field to conduct such interesting follow-up experiments.

Page 5; ACP interactions and aliphatic chain burial: Consider adding a reference to DOI: 10.1021/bi5014563

As suggested by Reviewer #1 we have now included this reference in our manuscript

Page 6, 7: The authors suggest that using data collected in 2.5 days and processed in a few days/2 weeks, their preparations of mycobacterial FAS-I with added commercial inhibitor should be suited to resolve structural details of inhibitor interactions. This indeed appears to

be a feasible and relevant experiment, and could be included in this manuscript to increase the overall relevance of the work.

We certainly intend to elucidate additional interesting structures, which would be the topics of additional future manuscripts. Our current limitation is getting sufficient microscope time for high-end data collection, a limitation that many structural biologists currently experience.

Figures:

Discussion of ACP linkage and orientation would require at least one figure with fitted ACP that allows assessment of relative orientation of ACP and proximal enzymatic domains, e.g. KS.

As we describe below in response to the comments of Reviewer #2, we refrain from discussing in the revised manuscript about the linker between ACP and MPT.

As for fitting ACP into the density that we have observed, this would be problematic due to a lack of suitable models.

The ACP from Mtb FAS-I is significantly different from all the other known ACP domains. Blasting its sequence against all the structures at the PDB does not provide any significant hit for this purpose.

A pairwise alignment of the Mtb FAS-I ACP with a fungal FAS-I ACP (see below) indicates identity that is too low for any reasonable modeling.

Thus, we cannot provide this kind of analysis.

```
#=====
#
# Aligned sequences: 2
# 1: MtbACP
# 2: FungalACP
# Matrix: EBLOSUM62
# Gap_penalty: 10.0
# Extend_penalty: 0.5
#
# Length: 242
# Identity:      45/242 (18.6%)
# Similarity:   77/242 (31.8%)
# Gaps:         89/242 (36.8%)
# Score: 83.5
#
#
#=====

MtbACP      1  PEPEPEEPEPVAESPAPDVVSEAAPVAPAASSAGPRPDDLVFDAADATL-      49
                                         : : : : : | : |
FungalACP   1  -----IADEPVKASLL      11

MtbACP      50  --ALIALSAKMRIDQIEELDSIESITDGASSRRNQLLVDLGSELNLGAID      97
      . : | . . . | . . . : | . . . : | . . . : | . . . : | . . . : | . . .
FungalACP   12  LHVLVAHKLKKSLSIPMSKTIKDLVGGKXTVQNEILGDLGKE--FGTTP      59

MtbACP      98  GAAESDLAGLRSQVTKLARTYKPYGPVLSDAINDQLRTVLGP--SGKRPG      145
      . . . | . . . : | . . . : | . . . : | . . . : | . . . : | . . . : |
FungalACP   60  EKPE-----ETPLEELAETFQ---DTFSGALGKQSSSLLSRLISSKMPG      100

MtbACP     146  ---AIAERVKKT-WELGEGWAKHVTVVEALGTREGSSVVRGGAMGHLHEG      190
      . : | . . . : | | . | . . . | . : | | | . . . . . : : |
FungalACP   101  GFTTITVARKYLQTRWGLPSGRQDGVLL-VALSNEPAARL-----G      139

MtbACP     191  ALADAASVDKVIDAAVASVAARQGVSVLPSAGSGGGATIDA      232
      . : | | | . : . : | : . . . . : | : . . : | | . :
FungalACP   140  SEADAKA---FLDSMAQKYASIVGVDL-----      163
```

Using a consistent colour scheme for domains across figures would be helpful.

We made an effort to keep the color scheme throughout the paper. Whenever the individual domains are shown we use the same color scheme as seen in figure 2. When comparing the Mtb and fungal FAS-I however we had to use a grey/pink color scheme as well as in some other instances. Otherwise, too many colors would be presented in a single figure or the colors of the model and of the electron density maps would be too similar, which would thus make it hard to interpret.

Ext. Data Fig. 2a: Please provide the full set of FSC curves produced by Relion.

As requested by reviewer #1, we included all the FSC curves that Relion outputs as well as a map-to-model correlation that was calculated using Phenix.

Ext. Data Fig. 4: Interpretation of central averaged density as individual ACP orientations would require improved data. It is difficult to exclude that the current maps still provide an ensemble average of multiple ACP positions.

See above our additional analysis for the ACP location and the additional Extended Data Fig 7 that we include.

Ext. Data Fig. 6: Indicate KS active site and a measurement of length of the path to the most plausible/possible PPant attachment point on ACP.

Since we cannot model ACP we cannot measure lengths and we do not know the path of the PPant. Nevertheless, we now added a scale bar to Extended data figure 6 (Fig. 8 in this revised manuscript) that is positioned next to the KS catalytic site (highlighted in red).

Ext. Data Fig. 8/9: Provide full methods for electrostatic potential calculation and a colour scale together with the figures.

In the initial version of the paper we presented surface electrostatic potentials in vacuum that were calculated by PyMol. In the revised manuscript we recalculated the electrostatic surfaces using APBS tools and we now present the electrostatic potentials at the solvent accessible surfaces mapped to the molecular surfaces and include a scale for the values as requested by Reviewer #1. We further describe the method for these calculations.

Ext. Data Fig. 11,13: Provide methods for tunnel calculation

We now explain in the methods section how we detected and visualized of internal cavities.

Reviewer #2 (Remarks to the Author):

The manuscript by Elad et al. describes the structure of the 2 MDa multienzyme complex fatty acid synthase (FAS) from Mycobacterium tuberculosis determined by cryo-electron microscopy. FAS is responsible for making the long fatty acids that constitute the mycobacterial cell wall, an important factor in virulence of M. tuberculosis. The complex was expressed in E. coli together with the phosphopantetheinyl transferase that is necessary to activate the acyl carrier protein (ACP) domain that carries the growing fatty acid chain between the different catalytic sites. The resulting preparation yielded a cryo-EM map with a

resolution of 3.3 Å, showing a rigid structure with heterogeneity only for the ACP inside the barrel. 3D classification revealed the ACP mostly docked at the ketoacyl synthase domain, as is also the case for eukaryotic FAS structures. Compared to the fungal FAS systems, the authors find larger catalytic clefts for some enzymes, compatible with the longer fatty acids produced by mycobacterial FAS. In addition, the authors observe a reversal of the electrostatic potentials of the catalytic domains.

The paper shows the first atomic resolution structure of mycobacterial FAS, an enzyme important for *M. tuberculosis* virulence and a potential drug target. For this reason, it will be of high interest in the field. The EM work is of high technical quality and the paper is overall well written; however, the presentation of the data needs some attention. The quality of the cryo-EM map is not shown clearly in any of the figures, so that the resolution claims can not be judged. In addition, details of the structure are difficult to interpret in still images and it would be useful to add some movies.

Specific points:

1. The last paragraph of the introduction describes the features and importance of the mycobacterial FAS system. It seems more logical to put this material before the section that describes the details of the structure.

As suggested by Reviewer #2, we now rearranged the introduction.

2. p. 4 first paragraph. The authors claim a resolution of "almost 2.5 Å" for parts of the structure. Such a claim should be supported by figures showing the relevant map regions with fitted model. At this resolution, most rotamers are unambiguous and water molecules will be visible. However, figure 1b does not really indicate much density beyond 3 Å. This means that the "unambiguous assignment of side chain rotamers for the vast majority of the amino acids" is also probably an overstatement. A supplementary figure with details for more regions than the very short helix in figure 1c should be included. (Resolution claims below 3 Å would be supported by side chains of isoleucines, and an obvious difference between phenylalanines and tyrosines.) A panel for the active site of each catalytic domain would be preferable. Also figure 1c should be replaced with a better example. The helix shown contains several methionines and their densities can't be judged from the angle that is shown.

We previously stated that the resolution at the core regions approach 2.5Å based on local resolution analysis. Since this is the upper resolution limit, only few local regions actually approach it. The more prevalent resolution in the core regions is better than 3.0Å (as clearly seen in figure 1b, dark blue and purple regions). To better illustrate the quality of the maps at the core regions we now added two additional figures. New Extended Data Figure 3 shows slices of the model and of the density map, giving an overview of all the six catalytic modules as requested by Reviewer #2. In addition, we added New Extended Data Figure 4 showing examples for the ability to determine the correct rotamer for isoleucine and to distinguish between phenylalanine and tyrosine residues based on electron density map. In addition, we also tuned down the text to avoid overstating the actual resolution achieved and the quality of our model. We now write "The local resolution of the map extends beyond 3.0 Å at core regions of the catalytic subunits and decreases to ~6 Å at the more peripheral and flexible regions (Fig. 1b). This map allowed us to model the structure of FAS-I including unambiguous assignment of side chain rotamers for most amino acids at the better resolved regions (Fig. 1c)."

Also, we now changed Figure 1c to present a strand with alanine, valine, leucine and asparagine residues.

3. p. 4, ACP classification: seeing multiple orientations of the ACP docked on the KS domain is interesting, but it should not be overinterpreted. The classification was done with D3 symmetry applied, but there is no reason to assume that the six ACP domains in a complex, or even the three in one dome, are communicating. This means that the observed structures are probably averages of different conformations, as also suggested by the low resolution of ~15 Å. However, the authors even claim to see the linker between the ACP and the MPT domain in one of the classes. Even if it was rigid, such a feature would not be visible at 15 Å resolution. The interpretation of this linker, including figure 2b and extended data figure 5, should be removed.

We accept the criticism of Reviewer #2.

Regarding the symmetry imposed on the ACP domains, we have now added localized reconstruction analysis as described by Ilca S. et al. Nature Communications 2015. Using this protocol individual ACP domains were extracted and 3D classified without imposing symmetry (Extended Data Figure 7).

Regarding the linker, given the low-resolution information that we have for this region we cannot confidently claim that the ACP connectivity is indeed as appears in the map. We do think that this density is not an artifact but we also acknowledge that the level of uncertainty makes our interpretation less convincing. We therefore removed Extended figure 5, modified figure 2b, and we no longer discuss the linker between the ACP to MPT in the manuscript, as requested by Reviewer #2.

4. p. 4, last paragraph: it is not clear if the KR loop is unique for mycobacteria or is also present in fungal FAS. This is relevant for the discussion of electrostatic surfaces.

Such long loop seems to be unique to mycobacteria (as could be seen in Figure 3). We now also added a clarification to the text.

5. p. 4/5 and Ext. data figure 7: the header of the chapter is "inverted potentials". But the figure (b vs. d) suggests local changes from positive to neutral and from neutral to negative, which means overall more negative charges, but locally not inverted. In both species, the area above the active site (arrow in b and d) is positive and below it negative.

Reviewer #2 is correct that some areas changed from neutral to negative and from positive to neutral but the overall effect is that the local electrostatic potential near the active site was inverted from positive to negative. As we mention above, we have recalculated the electrostatic potentials using the more precise APBS tools. In the new Extended Data figure 7 (Figure 9 in the revised manuscript) the inversion of the electrostatic potential is much more pronounced.

6. p. 5 bottom: "a marked diversity in the catalytic clefts of some modules". As only two species are considered, the authors probably mean "a marked difference". A similar issue, top of p. 6: "variations in amino acid composition".

We modified the text as Reviewer #2 suggested.

7. Figure 3: it is very hard to understand this figure, especially the surface representations. Movies or stereo figures would be better to give a 3D impression.

To make figure 3 easier to grasp, we now include supplementary movies that provide better 3D representations for the four catalytic domains.

8. Extended data figure 5: this figure is impossible to interpret and would be better shown as a movie or at least a stereo figure. But as stated in point 3 above, the conclusion about the connectivity of ACP and KS is not supported by data.

As explained above, we now removed Extended Data Figure 5 we no longer discuss the ACP – MPT linker.

9. Extended data figure 6: Also this figure is hard to interpret, in particular it is not clear how the KR loop model relates to the density map, and how it inserts between ACP and KS.

We now include an Extended Data movie showing the density of the ACP domain in respect to the KS module.

10. Table 1: Model to map correlation 0.8282: it is not clear what this one figure means. It would be better to add a map-to-model FSC curve to figure 2a. The map-to-model FSC 0.5 resolution should be similar to the half-map resolution at FSC 0.143.

As requested by Reviewer #2, we now add a model-to-map FSC curve to Extended Data figure 2a, and indeed it has an FSC=0.5 value at a resolution that is close to the half maps resolution at FSC=0.143.

11. Table 1: please add MolProbity score and clashscore.

We now include these values as requested.

REVIEWERS' COMMENTS:

Reviewer #2 (Remarks to the Author):

My previous comments have been satisfactorily addressed and the beautiful new supplementary figures with details of the map and the model support the high quality of the cryo-EM map. I recommend publication.